# Bioactivity and Chemical Profile of *Rubus idaeus* L. Leaves Steam-Distillation Extract

**DOI:** 10.3390/foods11101455

**Published:** 2022-05-17

**Authors:** Diana De Santis, Katya Carbone, Stefania Garzoli, Valentina Laghezza Masci, Giovanni Turchetti

**Affiliations:** 1Department for Innovation in the Biological, Agrofood and Forestry Systems, University of Tuscia Via S. C. de Lellis, 01100 Viterbo, Italy; laghezzamasci@unitus.it (V.L.M.); g.turchetti@unitus.it (G.T.); 2CREA Research Centre for Olive, Fruit and Citrus Crops, Via di Fioranello 52, 00134 Rome, Italy; katya.carbone@crea.gov.it; 3Department of Drug Chemistry and Technology, Sapienza University, Square Aldo Moro 5, 00185 Rome, Italy; stefania.garzoli@uniroma1.it

**Keywords:** raspberry leaves, steam-distillation, antibacterial activity, radical scavenging capacity, GC/MS profile

## Abstract

The leaves of *Rubus idaeus* L., a by-product of the fruit food industry, are a known source of bioactive molecules, although the chemical composition has only been partially investigated. The main objective of this study was to examine the biological activities and the chemical composition of the extract of leaves of *R. idaeus* (RH), obtained by steam distillation (SD). The antioxidant capacity; the total phenolic content (TPC); the cytotoxic activity against tumor cell lines; and the antibacterial activity, in addition to the study of the chemical fingerprinting, carried out by Gas/Chromatography-Mass-Spectrometry (GC/MS) and Headspace (HS)-GC/MS, were established. The extract showed a strong antioxidant capacity and a modest antibacterial activity against two bacterial strains, as well as significant cytotoxic activity against tumor cell lines (Caco-2 and HL60) and being proliferative on healthy cells. Many of the GC-identified volatile molecules (1,8-cineol, β-linalool, geraniol, caryophyllene, τ-muurolol, citral, α-terpineol, 3- carene, α-terpinen-7-al, etc.) can explain most of the biological properties exhibited by the extract of *R. idaeus* L. The high biological activity of the RH and the high compatibility with the various matrices suggest good prospects for this extract, both in the food and cosmetic fields or in dietary supplements for improving human health.

## 1. Introduction

*R. idaeus* L., called “Raspberry” or “Red Raspberry”, is a plant that belongs to the Rosaceae family, widely cultivated in Europe, North America, and Asia. Raspberry fruits are one of the world’s most consumed berries, available as fresh, frozen, and freeze-dried commodities. Moreover, they are also used in food manufacturing for purees, juices, jams, wines, etc. According to Faostat [1], in 2020, the world production of raspberries was 895.771 tons, with a cultivated area of just over 110,000 hectares, with Europe holding the highest share of raspberry production. These fruits are commonly known as superfoods because of their very high content of natural antioxidants and vitamins [2]. Besides, their production and processing generate different byproducts, such as stems, seeds, and leaves, which can be recovered and incorporated into new food and cosmetic or pharmaceutical products [3,4,5]. In particular, the therapeutic properties of the leaves of *R. idaeus* L. have been known since ancient times, recommended for the treatment of various unhealthy conditions, and included in herbal preparations for relaxation of the uterus during childbirth [6,7]. In addition, raspberry leaves are used as an additive in beverages and dietary supplements and in the formulation of functional herbal teas, teas, and chocolate, improving their nutritional properties and flavor [7,8,9,10,11,12,13]. 

Antibiotic resistance is a health problem that causes the deaths of around 700,000 people every year, and the scientific world predicts it will rise to 10 million a year by 2050 [14]. Bacteria acquire resistance to antibacterial agents according to two possible mechanisms, vertical (by the isolated evolution of the strain) or horizontal (by the resistance genes exchanged between strains or individuals of the same species). World trade; population growth; and the massive use of antibiotics in agriculture, fisheries, and veterinary medicine raise the issue of the multi-drug resistance of microorganisms in the food chain [15,16,17]. In addition to the above, Mensah and colleagues [18] pointed out that residues of antimicrobial agents in food can lead to public health problems due to the emergence of multidrug-resistant strains, allergic reactions, hepatotoxicity, and other medical complications. The unsuccessful and irresponsible use of current antibiotics is becoming a growing problem. For this reason, the study of active plant matrices has focused on the search for molecules that can exert an antimicrobial action alone or in combination with existing antibiotics. To the best of our knowledge, detailed research of raspberry leaves distillate has not yet been undertaken.

In this regard, the volatile compounds of *R. idaeus* leaves were isolated by steam distillation, and the obtained distillate was subsequently analyzed by HS-GC/MS and GC/MS. To further evaluate the potential of the distillate for applications in food, medicine, and health products, its total polyphenol content, as well as antioxidant capacity and antiproliferative and antibacterial activities were assessed in vitro towards DPPH, ABTS, FRAP, human colorectal adenocarcinoma cells, human promyelocytic leukemia cells, normal breast epithelial cells, and different strains of both Gram positive and negative bacteria, respectively.

## 2. Materials and Methods

### 2.1. Reagents and Chemicals

All the chemicals used for the experiments were of analytical grade. Folin–Ciocalteau, sodium carbonate, gallic acid, reagents for the cell and bacteria cultures, and antiproliferative antibacterial assays were purchased from Merck KGaA (Darmstadt, Germany); 2,2′-azino-bis (3-ethylbenzothiazoline-6-sulphonic acid) (ABTS), 2,2-diphenyl-1-pikryl-hydrazyl (DPPH), and other antioxidant reagents were obtained from Sigma (St Louis, MO, USA). Dimethyl sulfoxide was purchased from Biochem Chemopharma (Cosne-Cours-sur-Loire, France) and gentamicin from Biochrom AG (Berlin, Germany).

### 2.2. Plant Material, Experimental Design, and Extraction Process

Raspberry (*R. idaeus* L.) leaves, cv. *Heritage*, used in this study were harvested in September 2020, at the “Lamponi Monti Cimini” farm, located in Viterbo (Lazio region, Italy), at about 750 m above sea level, with a moisture content of approximately 31 ± 1% (*w*/*w*). The leaves, immediately transported to V. Cardarelli Institute laboratory (Tarquinia, Viterbo, Italy) after harvesting, were weighed, washed, and steam-distilled using a 30-L extractor from Manufattinox company (Scambiatterra Francesco—Varapodio, RC, Italy). In the distillation process, about 2 kg of fresh raspberry leaves were placed inside the boiler, on a perforated grid, positioned about 3 cm above 3.5 L of boiling water, and a hydrolate (RH) was collected (200 mL) and stored at 4 °C until extraction.

### 2.3. Moisture

The leaf moisture value was determined on an aliquot of dried sample (about 5 g) in an oven (NF 400 Nüve Sanayi, Akyurt-Ankara) at 65 ± 2 °C until a constant weight was reached. Three replicates were performed, and the results were averaged.

### 2.4. Total Phenol Content Assay

Total Phenol Content (TPC) of RH samples was evaluated spectrophotometrically according to the Folin method [19]. Briefly, 20 µL of RH, diluted with 1580 µL of distilled water and 100 µL of Folin—Ciocalteu reagent (2 M), was incubated for 5 min in the dark, at room temperature. Afterwards, 300 µL of Na_2_CO_3_ (20%, *w*/*v*) was added, and the mixture was left at room temperature for 2 h. A calibration curve was generated with standard solutions of gallic acid (20–200 µg mL^−1^), and the measures were carried out at 760 nm using a UV-VIS spectrophotometer (Perkin Elmer model Lambda 850+). All analyses were performed in triplicate. TPC was expressed as milligrams of gallic acid equivalents per mL of sample (mg GAE mL^−1^).

### 2.5. VOCs Analysis

#### 2.5.1. Headspace-Gas Chromatography Mass Spectrometry (HS-GC/MS)

A Perkin–Elmer Headspace Turbomatrix 40 autosampler connected to a Clarus 500 GC/MS was used for the chemical qualitative–quantitative analysis of the vapor phase of RH. A total of 2 mL of the sample was placed in a 20 mL vial sealed with headspace PTFE-coated silicone rubber septa and caps. The headspace-applied parameters followed Garzoli et al. [20], with some modifications. The GC was equipped with a Restek Stabilwax (fused-silica) polar capillary column, and helium as carrier gas (1.0 mL min^−1^) was used. The column temperature was programmed as follows: from 60 °C to 220 °C at a rate of 5 °C min^−1^ and held for 15 min. The MS parameters were ionization voltage taken at 70 eV; mass range was from 40 to 500 *m*/*z*, ion source temperature of 200 °C, and scan time of 0.2 s.

Relative percentages for quantification of the components were calculated by electronic integration of the GC/FID peak areas using the normalization method without using corrections factors (RRFs). The identification of the components separated by GC/MS was performed first by comparing the mass spectra for each compound with that reported on the MS libraries database (Wiley and Nist 02) and then by comparison of Linear retention indices (LRI) of each compound, calculated using a mixture of n-alkanes (C8-C30, Ultrasci), with those reported in the literature for an apolar column. All analyses were performed in triplicate.

#### 2.5.2. Gas Chromatography Mass Spectrometry (GC/MS) of Hexanoic Extract

For GC/MS analysis, because no essential oil was isolated, volatile compounds were extracted from the hydrosol obtained by steam distillation. Briefly, 20 mL of RH were mixed with 1 mL of n-hexane and stirred in the dark at room temperature. Subsequently, the hexane layer was collected and stored at −20 °C until use. The gas chromatography–mass spectrometry (GC/MS) analysis was performed on an Agilent 7890A Series GC system, coupled with a 5975C mass detector (Agilent Technologies, Milan, Italy). An Agilent Ultra Inert GC column with an HP-5MS-fused silica capillary (5%-diphenyl95%-dimethyl polysiloxane, 30 m, 0.25 mm i.d., 0.25 mm film thickness) (Agilent Technologies, Palo Alto, CA) was used to provide the analyte separation, and helium was used as a carrier gas (1 mL min^−1^). The GC oven temperature gradient started from 50 °C and was held for 3 min; then, the temperature was raised to 200 °C (5 °C min^−1^). The final temperature was maintained for 18 min [21]. The injector was maintained at 250 °C, operating in the spitless modality.

The mass spectrometer was equipped with an electron impact (EI) source (70 eV), and the acquisition mode was in full scan (from 40 to 600 *m*/*z*). A solvent delay time of 4.10 min was applied. GC/MS data were acquired, and the total ion chromatograms (TIC) were integrated by using MassHunter software (version B.05.00; Agilent Technologies, Milan, Italy). Quantitative analyses of each component (expressed as area percentage) were carried out by a peak area normalization measurement, calculated as mean values of three sample injections. The linear retention index (LRI) of each compound was calculated, referring to the retention times (RT) of a C7-C25 n-alkanes standard mixture under the same conditions.

The identification of the compounds was performed by matching their recorded mass spectra with the standard mass spectra in NIST11 library and in the National Institute of Standards and Technology Gas Chromatography Library (http://webbook.nist.gov/chemistry, accessed on 16 July 2020), as well as by comparing their MS fragmentation patterns and LRIs with literature data [22,23,24] and the available online mass spectra data bank (MoNA—MassBank of North America; https://mona.fiehnlab.ucdavis.edu/, accessed on 16 July 2020).

### 2.6. Cell Viability Assay

#### 2.6.1. Eukaryotic Cell Culture

Three different cell lines were used: human colorectal adenocarcinoma cells (Caco-2, ATCC^®^ HTB-37™), human acute promyelocytic leukemia cells (HL60, ATCC CCL-240), and normal breast epithelial cells (MCF10A, ATCC^®^ CRL-10317™). Caco-2 cells were cultured in high glucose Dulbecco’s Modified Eagle Medium (DMEM), supplemented with 10% Fetal Bovine Serum (FBS), 1% *w*/*v* glutamine, 1% *w*/*v* sodium pyruvate, 1% *w*/*v* non-essential amino acids, and 1% penicillin-streptomycin. HL60 were cultured in Roswell Park Memorial Institute (RPMI)-1640, supplemented with 10% FBS (*v*/*v*), 1% glutamine (*w*/*v*), and 1% penicillin-streptomycin (*v*/*v*). MCF10A were maintained in DMEM F-12 supplemented with 100 ng mL^−^^1^ cholera toxin, 20 ng mL^−^^1^ epidermal growth factor (EGF), 0.01 mg mL^−^^1^ insulin, 500 ng mL^−^^1^ hydrocortisone, 5% Horse Serum (HS) (*v*/*v*), 1% glutamine, and 1% penicillin-streptomycin; before being seeded for the viability assay, the culture medium was deprived of Epidermal Growth Factor (EGF) and the HS reduced to 2%. All the cell lines were maintained at 37 °C in a humidified 5% CO_2_ atmosphere condition. All experiments were performed with the cells in their logarithmic growth phase.

#### 2.6.2. Cytotoxicity Assay

In vitro cytotoxic effects of RH were investigated in triplicate by the MTT (3-(4,5-Dimethylthiazol-2-yl)-2,5-Diphenyltetrazolium Bromide) assay, according to Nguyen and colleagues [25] with minor modifications. Briefly, on 2 × 10^4^ cells/well seeded in a 96-well microplate 24 h before treatments, 10 2-fold dilutions from 25% to 0.05% (*v*/*v*, Medium/RH) were used and incubated for 48 h. After the incubation time, the medium-containing treatment was removed, and 100 µL of MTT solution (0.5 mg mL^−^^1^) was added to each well and incubated in the dark at 37 °C for 3 h. The formazan crystals were dissolved in 100 µL of DMSO, and the absorbance was measured at 570 nm. The cytotoxicity effect of RH was estimated as the percentage of viable cells relative to a control (untreated cells).

### 2.7. Antibacterial Assay

#### 2.7.1. Prokaryotic Cell Culture

Five bacteria strains were used to evaluate the antibacterial effects of RH: three Gram-negative strains (*Escherichia coli* ATCC 25922, *Acinetobacter bohemicus* DSM 102855, and *Pseudomonas fluorescens* ATCC 13525) and two Gram-positive strains (*Bacillus cereus* ATCC 10876 and *Kocuria marina* DSM 16420). All strains were maintained in Lysogeny agar at different temperatures: 26 °C for *B. cereus, P. fluorescens*, and *A. bohemicus* and 37 °C for *K. marina* and *E. coli*. All inocula were prepared with fresh bacteria plated the day before the test.

#### 2.7.2. Agar Diffusion Method

The antimicrobial activity of samples analyzed was investigated by the disk diffusion test, according to Hudzicki [26]. Each bacterial strain was tested at 0.5 McFarland, which is equivalent to a bacterial suspension containing between 1 × 10^8^ and 2 × 10^8^ CFU mL^−1^, in duplicate, and the halo of the inhibition zones was measured using a vernier caliper rule.

### 2.8. Antioxidant Activities

#### 2.8.1. DPPH^•^ Scavenging Activity

The DPPH radical scavenging activity was determinate according to the protocol described by Blois [27], adapted for 96-well plates. A total of 100 µL of different concentrations (1000–0.49 µM) of Trolox (6-hydroxy-2,5,7,8-tetramethylchroman-2-carboxylic acid) and 100 µL of RH were mixed and serially diluted (1:2) in methanol. Then, 100 µL of 0.2 mM DPPH• pure methanol solution was added to each well and kept at room temperature for 30 min in the dark. After, the absorbance was read at 517 nm in a UV spectrophotometer plate-reader (Sunrise, Tecan, Inc., San Jose, CA, USA). The results were expressed as the µM of Trolox Equivalents (TE) mL^−^^1^ of RH used.

#### 2.8.2. ABTS^•+^ Scavenging Activity

The antioxidant capacity was assessed by an ABTS assay as described by Bueno-Costa et al. [28], with minor modifications. Briefly, 5 mL of 7 mM ABTS in distillated water was prepared and mixed with 88 µL of 140 mM potassium persulfate (K_2_S_2_O_8_) and left to react for 12–16 h at room temperature under dark conditions. The ABTS^•+^ solution was diluted with ethanol until an absorbance of 0.7 ± 0.02 was read at 734 nm prior to use. Trolox solution was prepared and used as a standard at different concentrations (0.25–4 µM). A total of 20 µL of sample/standard were added to 980 µL of ABTS^•+^ solution and left to react for 5 min. After the reaction time, the absorbance was evaluated using a spectrophotometer (Perkin Elmer Lambda 25 UV/VIS spectrometer, Norwalk, CT, USA) at 734 nm. The results were expressed as µM of TE mL^−1^ of RH used.

#### 2.8.3. FRAP Assay

The procedure described by Gül and Pehlivan was followed [29]. The principle of this method is based on the reduction of a ferric-tripyridyl triazine complex to its ferrous, colored form in the presence of antioxidants. The results were expressed as µM of TE mL^−^^1^ of RH used.

### 2.9. Statistical Analysis

Data were reported as mean ± standard deviation (SD) of at least two independent experiments with three replicates. A one-way analysis of variance (ANOVA) was used to analyze the data, followed by the Least Significant Difference (LSD) test (*p* < 0.05) (XL-Stat-Addinsoft, 2019)

## 3. Results

### 3.1. General Aspects

In the present study, steam distillation and extraction processes were applied to fresh raspberry leaves, cv Heritage, to produce and characterize a functional distillate to be used in sectors like food, cosmetics, and phytotherapic ones. Steam distillation allows a good separation of the plant’s volatile active components, avoiding the solvent removal phase or the using of aggressive chemicals, which could create artifacts in the distillate. In the experimental conditions used, a 10% (*v*/*w*) distillate was obtained, with a pH value 4.06 ± 0.05 and a density of 0.993 ± 0.004 g cm^−^^3^.

Raspberry leaves are rich in bioactive compounds, mainly hydrolysable tannins (2.6–6.9% of dried leaf mass); flavonoids, including quercetin and kaempferol-3-O-glucoside (0.46–1.05%) [30]; and smaller amounts of phenolic acids, such as chlorogenic, gallic, ferulic, and caffeic acids [31].

However, the knowledge of the composition and the positive effects on health of the bioactive substances contained in the distillate of raspberry leaves is not as widespread as the knowledge about their polyphenolic profile.

Although no reports on volatile metabolites from *R. ideaus* leaves have been published, other Rubus species have been phytochemically investigated for their volatile compounds. Cai et al. [32] analyzed the components contained in the hydro-distillated volatile oil from *R. parvifolius* L. leaves by GC/MS. They identified 29 compounds, with 4-hydroxy-3-metoxystirene being the most abundant one (66.05%), followed by 2-hexadecen-1-ol (9.79%), 4-tert-butyl benzoic acid (2.22%), hexanoic acid (2.03%), and linalool (1.39%). The authors also pointed out that the extracted oil effectively inhibited the growth of a wide range of Gram positive and negative bacteria. The importance of discovering new antibacterial agents is growing every year.

### 3.2. HS-GC/MS Identification of Volatile Compounds

The volatile organic compounds (VOCs) in the headspace of the RH were analyzed by HS-GC/MS analysis. Table 1 lists all the identified compounds. In the RH headspace, 62.6% of the compounds found were oxygenated monoterpenes, mainly 1,8-cineole (50.8%), α-terpineol (5.2%), terpinyl acetate (3.7%), and camphor (2.9%).

Among the monoterpene hydrocarbons, 3-carene (16.3%) was the most abundant, followed by limonene (1.0%) and β-mircene (0.1%). Monoterpenes are well-known components of essential oils of aromatic plants, are able to penetrate the bloodstream, and act as medicinal substances, beneficial for humans [33,34,35,36].

VOCs found in the RH headspace have important biological properties, as reported in the literature data. Among these, 1,8-cineole stands out in percentage terms, probably for its high solubility in water. This compound, found in several species of aromatic plants (eucalyptus, thyme, rosemary, sage, etc.), is a cyclic ether monoterpenoid naturally produced, approved by the Food and Drug Administration for food use, and, in 1965, it obtained GRAS (Generally Recognized As Safe) status by FEMA (Flavor and Extract Manufacturers Association of the United States). Its pharmacological properties have attracted the attention of the scientific world. It is an anti-inflammatory of the airways, a mucolytic agent, and a powerful antibacterial agent against numerous strains, such as *Enterococcus faecalis*, *Streptococcus salivarius*, *S. sanguinis*, *B. subtilis*, *Staphylococcus aureus, E. coli*, *S. epidermidis,* and *Aspergillus niger* [37], as well as in vitro and in vivo antitumor effects [38,39].

3-carene is a bicyclic monoterpene produced by the cyclization of geranyl diphosphate, with several properties, such as antimicrobial, antioxidant, anti-inflammatory, antiplasmodial, cytotoxic, and anti-tuberculosis properties [40].

Worthy of attention is the presence in the RH of α-terpineol (5.2%), an unsaturated monocyclic monoterpene alcohol, isolated from leaves, flowers, and aerial parts of plants. It is usually a mixture of isomers (α, β, γ-terpineol and terpin-4-ol), with α-terpineol as the main constituent. α-terpineol has a pleasant lilac-like odor and is a common ingredient in perfumes, cosmetics, and flavorings. It is also well known for its good antibacterial activity [34] and various important biological properties [41].

Among the active components of our RH, camphor (2.9%) was detected, which is also one of the main components of many plant species, with documented antibacterial and antimicrobial properties.

Limonene (1%) has antibacterial, antifungal, anti-inflammatory, antioxidant, and tumor-suppressive activity [42,43,44,45,46].

### 3.3. GC/MS Identification of Hexanoic Extract Compounds

The distillate collected after the steam distillation was extracted with hexane to recover the volatile compounds dissolved in the hydrosol. The GC/MS analysis of the extract revealed the presence of 80 phytochemical compounds (Figure 1), some of them not detected through HS-GC/MS analysis. The RH was rich mainly in aldehydes, ketones, alcohols, and terpenoids.

The main compounds (>1% of the total peak area) identified, characterizing the sample composition, are reported in Table 2. The major constituents of the volatile hexanoic fraction of the hydrolate were 2-hexenal (8.79%); 2-nonanone (6.88%), followed by some monoterpenoids; and 2,4-heptadienal (3.13%), among the known compounds.

Both α and β-unsaturated aldehydes and long chain methyl ketones are known to exert antimicrobial and antifungal activity [47,48,49,50,51,52,53]. The main terpenoids detected were β-linalool (6.15%), geraniol (4.45%), 1,8-cineole (2.47%), τ-muurolol (1.09%), and α-citral (1.00%) and the sesquiterpene β-caryophyllene (2.61%).

Furthermore, several literature studies have underlined the biological activity of linalool, such as anti-inflammatory and antibacterial activity [54,55]. In addition, the hexane extract confirmed the presence of 1,8-cineole (2.47%), albeit at a much lower level than found in the headspace analysis.

The monoterpenoid α-terpinen-7-al was identified (1.19%), previously reported as a safe aroma compound [56]. Among the minor components (<1%), the following were identified: camphor (0.95%) and α-terpineol (0.92%), both also found in the HS-GC/MS analysis of the hydrolate.

Regarding methyl salicylate (1.94%), it is known for its antioxidant, antimicrobial, insecticidal, and acaricidal activities [57,58,59,60]. Furthermore, the analysis revealed the presence of two acids, lauric acid (dodecanoic acid: 1.37%) and capric acid (n-decanoic acid: 1.15%), both studied for their antifungal and antibacterial activity [61,62,63].

### 3.4. Biological Activities of Raspberry Leaf Distillate

#### 3.4.1. Cell Viability

Cancer therapies require continuous research to find alternative molecules for anti-tumor treatment. Over time, numerous phenomena of resistance to chemotherapy drugs have been highlighted by the scientific community, as reported by several studies, and it is thought to be the cause of mortality in more than 90% of patients with advanced cancer [64]. We wanted to test whether Rubus leaf hydrolate could have an anti-proliferative effect on cell cultures in vitro.

In the present study, the effects of the RH supplementation of the culture media were tested on two tumor cell lines, HL60 and Caco2, and on a healthy one, MCF10A. As reported in Figure 2, it is possible to notice how HL60, promyeoloblasts isolated from the peripheral blood by leukopheresis from a 36-year-old caucasian female with acute promyelocytic leukemia, and Caco2, epithelial cells isolated from colon tissue derived from a 72-year-old white male with colorectal adenocarcinoma, decreased in viability when treated with RH. Both HL60 and Caco2 cell lines showed a significant reduction (*p* < 0.05) in viability when treated with a 25% hydrolate solution. In particular, the residual viability of the two cell lines, HL60 and Caco2, were, respectively, 1.15% and 0.02%.

This result could be related to the presence, in the extract, of molecules such as terpenes with antitumor activity, including 1,8-cineole [65], limonene [42], α-terpineol [66], β-linalool [67], geraniol [68], caryophyllene [69], and τ-muurolol [70].

The trend observed regarding MCF10A, an epithelial cell line isolated in 1984 from the mammary gland of a 36-year-old white woman with fibrocystic breasts and used by researchers as a healthy cell model [71,72], was opposite to that related to tumor lines.

The cells showed an increase in viability when the medium was enriched with a solution of RH already at the concentration of 0.098%. When treated with 25% RH, these cells achieved a viability increase of up to 124.83%.

The antiproliferative activity of the active components of RH, identified by GC analyses, was exerted according to different mechanisms, as shown in the scientific literature.

The apoptosis is the most frequent action mechanism, common to 1,8-cineole, β-linalool, geraniol, caryophyllene, τ-muurolol, citral, α-terpineol, and 3-carene [38,73,74,75,76]. For the 1,8-cineole, specifically, the mechanism of action, partially clarified by Murata and colleagues [39], involves the activation of the caspase-3-dependent apoptosis pathway.

Other authors [65] have studied the mechanisms of action of 1,8-cineole, which confirmed cell cycle arrest in HepG2 hepatocellular carcinoma cells.

Furthermore, the antitumor activity could be related to the co-presence of 3-carene, observed by Yang and colleagues [77], on human lung cancer A549 cells. The anti-cancer activity was also demonstrated for α-terpineol, and the mechanism of action was partially elucidated by suppression of the NF-κB signal [41,78].

#### 3.4.2. Antibacterial Activity

Several bacterial strains were tested, known to be food contaminants (*E. coli, P. fluorescens*, and *B. cereus*) and of public health concern (*K. marina* and *A. bohemicus*). Regarding the latter, *K. marina* is a pathogen-emerging bacterium [79,80] which can contribute to the formation of biofilms in different stages of the food chain, increasing the chances that more pathogenic bacteria, such as *Listeria monocytogenes*, can survive the sanitation processes or carry out cross-contamination [81]. *A. Bohemicus*, which belongs to the genus Acinetobacter, is instead widely studied for its strong ability to generate multi-pharmacological resistance and the horizontal transmission of resistance-associated genes [82]. As reported by several authors, this bacterium can grow in the environment (soil and water), and its presence in the food chain can occur [83,84].

The antibacterial activity of the RH samples is reported in Table 3, and it is also presented in comparison with the reference drug gentamicin. The results showed a weak intrinsic antibacterial activity compared to the reference drug only against two bacterial strains: the *B. cereus* strain, for which an inhibition halo of 7.67 mm was recorded, and a slightly higher effect on *A. bohemicus*, for which an inhibition halo of 12 mm was recorded. No inhibitory effect on growth for the other bacterial strains (*E. coli, K. marina*, and *P. fluorescens*) was found. Growth inhibition by RH samples in the two susceptible strains was one-third of that exerted by the antibiotic gentamicin. These results, considering the dilution and the aqueous nature of the tested distillate, suggest that an appropriate concentration of extract could result in similar activity, reducing the risk of induced resistance, resulting in interest for a wide range of applications.

As some authors hypothesize, the antimicrobial activity of Rubus leaves extracts could probably be related to the presence of the phenolic components, particularly ellagic acid [85]. However, a reasonable explanation for the observed antibacterial activity could also be related to monoterpenes, such as 1,8-cineole, α-terpineol, limonene, and β-myrcene [36], identified in RH samples.

#### 3.4.3. TPC and Antioxidant Activity

To explore the intrinsic antioxidant capacity of raspberry leaf distillate, samples were tested for their ability to inhibit synthetic radicals in vitro. Results are reported in Table 4.

The value obtained in the DPPH assay was 6746 ± 555 µM TE 100 mL^−1^ of RH used, which is not very different from the values found in the literature [5,86,87,88,89]. Regarding the ABTS assay, the value obtained was 13.57 ± 0.56 µM TE 100 mL^−1^ of RH, which is slightly lower but still in line with that observed by other researchers [5,86,87,88,89,90,91,92,93,94]. In the FRAP assay, the value obtained was 307.48 ± 3.08 µM TE 100 mL^−1^ of RH. In this case, comparing the values with those found in the bibliography showed that the effects of the hydrolate in this assay were, in general, greater than other work [88,92,93,94,95,96,97,98,99,100]. The content of total phenols in the hydrolate was very scarce (6.25 ± 0.54 mg GAE 100 mL^−1^), consistent with the extraction technique adopted, and much lower than the extracts with the solvent [99,100,101,102,103,104,105].

The values of the antioxidant activity of Rubus, reported in many scientific works, show wide fluctuations. To compare the amounts found with other studies conducted on Rubus cultivars, we collected a large amount of information in Appendix A, reported in the Appendix A section. As is evident, even the values obtained with the same analytical method were affected by high variability as a function of numerous factors, such as the part of the plant used (fruit, leaf, seed), the extraction system (maceration, infusion, distillation, etc.), and the solvent used, as well as the cultivation system and the environment and agrometeorological condition. All of this is consistent with the natural variability in the biosynthesis of secondary metabolites as a plant response to external stimuli.

The low quantity of phenols in the Rubus hydrolate leads to the presumption that the antioxidant activity recorded depends on other non-phenolic components, such as 1,8-cineole, β-linalool, geraniol, caryophyllene, τ-muurolol, and limonene [106,107,108,109,110].

## 4. Conclusions

The integrated approach conducted in this study revealed some interesting biological properties of the raspberry leaves extract, obtained by steam distillation, without organic solvents and the related problems. The volatile fraction identified with HS-GC/MS was particularly rich in terpenoids, such as 1,8-cineole (50.8%), 3-carene (16.3%), 2-heptanol (10.3%), α-terpineol (5.2%), terpinyl acetate (3.7%), pentanal (2.9%), camphor (2.9%), 1-butanol, 2-methyl- (1.6%), 3-hexen-1-ol (1.5%), and limonene (1%), molecules known for their biological properties, and other minor components, which, by synergistic effect, could enhance their activity, providing a possible explanation for the multiple functional properties. On the other hand, the GC/MS analysis of the hexane extract of Rubus leaf hydrolate revealed other compounds, such as 2-hexenal (8.79%), 2-nonanone (6.88%), β-linalool (6.15%), geraniol (4.45%), 2,4 heptadienal (3.13%), caryophyllene (2.61%), 1,8-cineole (2.47%), methyl salicylate (1.94%), 2-hexanol-3-methyl (1.59%), 4-heptanol-3-ethyl (1.56%), 2-heptanone (1.49%), dodecanoic acid (1.37%), 3-hexanol-5-methyl (1.26%), n-decanoic (1.15%), α-terpinen-7-al (1.19%), τ-muurolol (1.09%), and citral (1.00%).

The biological action exerted by the *R. idaeus* leaf hydrolate was observed on *B. cereus* and *A. bohemicus* strains, with an inhibition halo of 7.67 ± 1 mm and 12 ± 3 mm, respectively.

In addition, selective cytotoxicity towards cancer cells (HL60 and Caco-2) was recorded, highlighting how dose-dependent this activity is. In contrast, the healthy cells tested (MCF10A) showed increased proliferation when treated with RH.

Therefore, the study demonstrates that thanks to the chemical composition, raspberry leaves are a by-product that could have an optimal use in the food, cosmetic and pharmacological industries, both to improve the preservation and stability capacities and the functional properties of potential derivatives.

## Figures and Tables

**Figure 1 foods-11-01455-f001:**
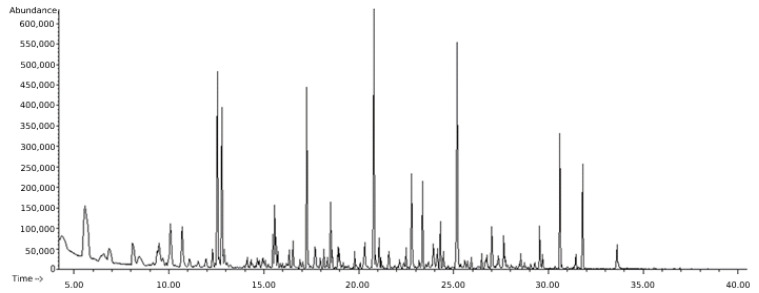
Total ion chromatogram (TIC) of a hexan extract of raspberry leaves steam-distilled.

**Figure 2 foods-11-01455-f002:**
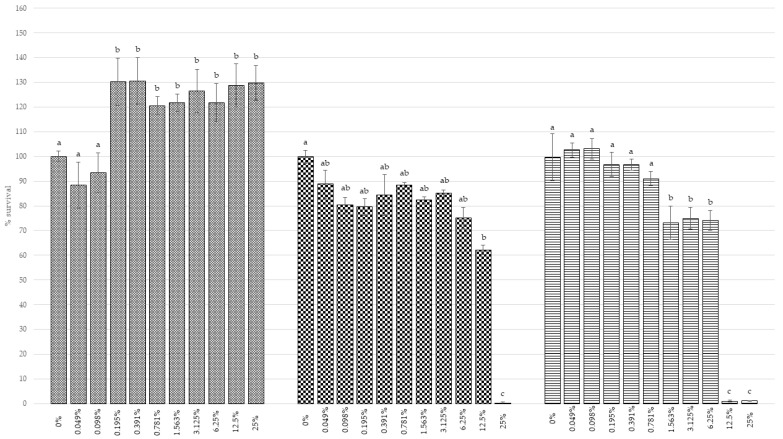
Antiproliferative activity after 48 h of treatment with RH tested on MCF10A, HL60, and Caco2 cell lines. The standard deviation is represented by black bars. Bars with different letters indicate a significant difference at *p* < 0.05 among cell lines, according to a one-way analysis of variance (ANOVA) for multiple comparisons applying the Tukey test.

**Table 1 foods-11-01455-t001:** Total volatile organic compound (VOC) profiles for the distillate of raspberry leaves by HS-GC/MS. LRI: Linear Retention Indices measured on a polar column; LRI^lit^: Linear Retention Indices from literature.

Peak	LRI	LRI^lit^	Component	Class	Amount (%)
1	915	914	butanal, 3-methyl	*Aldehyde*	0.2
2	970	968	pentanal	*Aldehyde*	2.9
3	995	993	2,3-butanedione	*Ketone*	0.4
4	1025	1022	2-butanol	*Alcohol*	0.4
5	1093	1095	1-propanol, 2-methyl-	*Alcohol*	0.4
6	1148	1146	3-carene	*Monoterpene*	16.3
7	1160	1157	β-myrcene	*Monoterpene*	0.1
8	1200	1198	limonene	*Monoterpene*	1.0
9	1210	1207	1-butanol, 2-methyl-	*Alcohol*	1.6
10	1214	1209	1,8-cineole	*Oxygenated monoterpene*	50.8
11	1298	1296	2-heptanol	*Alcohol*	10.3
12	1395	1389	3-hexen-1-ol	*Alcohol*	1.5
13	1398	1394	2-hexen-1-ol	*Alcohol*	0.2
14	1472	1469	5-hepten-2-ol, 6-methyl-	*Alcohol*	0.9
15	1531	1528	camphor	*Oxygenated monoterpene*	2.9
16	1585	1586	hotrienol	*Alcohol*	0.2
17	1618	1619	terpinyl acetate	*Oxygenated monoterpene*	3.7
18	1676	1675	α-terpineol	*Oxygenated monoterpene*	5.2
19	2014	2011	methyl eugenol	*Phenylpropanoid*	0.5
	Total				99.0

**Table 2 foods-11-01455-t002:** Volatile fraction composition of raspberry leaves hydrolate hexanoic extract determined by GC/MS. ^1^ LRI: Linear Retention Indices.

Peak	LRI ^1^	Compound	Class	Amount (%)
1	829	2-hexenal	*Aldehyde*	8.79
2	866	2-heptanone	*Ketone*	1.49
3	876	2-hexanol-3-methyl	*Alcohol*	1.59
4	914	4-heptanol-3-ethyl	*Alcohol*	1.56
5	924	3-hexanol-5-methyl	*Alcohol*	1.26
6	969	2,4 heptadienal	*Aldehyde*	3.13
7	984	1,8-cineole	*Oxygenated monoterpene*	2.47
8	1029	2-nonanone	*Ketone*	6.88
9	1035	β-linalool	*Oxygenated monoterpene*	6.15
10	1135	geraniol	*Oxygenated monoterpene*	4.45
11	1145	citral	*Terpenoid*	1.00
12	1162	unknown	*-*	1.68
13	1195	methyl salicylate	*Ester*	1.94
14	1198	n-decanoic acid	*Acid*	1.15
15	1210	unknown	*-*	6.71
16	1256	unknown	*-*	2.80
17	1269	unknown	*-*	2.11
18	1281	α-terpinen-7-al	*Monoterpenoid*	1.19
19	1289	unknown	*-*	1.15
20	1308	unknown	*-*	6.65
21	1352	dodecanoic acid	*Acid*	1.37
22	1410	τ-muurolol	*Sesquiterpene*	1.09
23	1437	unknown	*-*	3.30
24	1466	caryophyllene	*Sesquiterpene*	2.61
Total identified				72.52

**Table 3 foods-11-01455-t003:** Antibacterial activity of 10 µL of pure RH and 10 µL of Gentamicin (1 mg/mL) evaluated using the agar diffusion method. NE: No Effect.

Agar Diffusion Method	RH (10 µL) Inhibition Halo (mm) ± sd (mm)	Gentamicin (10 µL) Inhibition Halo (mm) ± sd (mm)
*E. coli*	NE	24.8 ± 0.4
*K. marina*	NE	26 ± 6
*P. fluorescens*	NE	22 ± 1
*B. cereus*	7.67 ± 1	22 ± 3
*A. bohemicus*	12 ± 3	34 ± 2

**Table 4 foods-11-01455-t004:** Total Phenols Content (TPC), expressed as mg GAE 100 mL^−1^, and the antioxidant capacity of RH in three different tests: DPPH, ABTS, and FRAP. Antioxidant values are expressed as µM of Trolox Equivalent 100 mL^−1^ of RH used in the assay.

	TPC	DPPH	ABTS	FRAP
**RH**	6.25 ± 0.54	6746 ± 555	13.57 ± 0.56	307.48 ± 3.08

## Data Availability

Data is contained within the article or Appendix A.

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
