# Peer review of "Bioactivity and Chemical Profile of Rubus idaeus L. Leaves Steam-Distillation Extract"

_foods, 2022, doi:10.3390/foods11101455_

Round 1
Reviewer 1 Report
The subject of this research is the red raspberry (Rubus idaeus L.) harvested in a farm in Lazio region, Italy. The aim of the study is the determination of biological activities and the chemical composition of leaves extract from R. idaeus. То achieve the research goal, the team used modern analytical techniques for preparative apply, e.g. steam distillation, and modern analytical techniques for quantification as UV-VIS spectroscopy, Gas/Chromatography-Mass-Spectrometry and Headspace-GC-MS. The determination of the cytotoxic activity against tumor cell lines, and the antibacterial activity of the extract obtained, has practical significance.
The topic and the goal are oriented ecologically due to the search for new sources of biologically active substances from the bio laboratory "Nature". Moreover, the object of the study is a cultivated plant species, classified as “super food”.
A reasonable solution is offered by the authors: the Rubus idaeus L are rich source of bioactive compounds with cytotoxic and antibacterial activities. The references cited are relevant, and more than half of them are from the last 10 years.
I have some questions and suggestions:
General:
- The main text doesn’t prepared according to the requirements of manuscript-writing-guidelines of the journal.
- It is not clear, how much samples are analyzed? How much are the repetitions of each samples?
Introduction:
- Line 44-58: In my opinion, the text there belongs to the Discussion section. The Introduction is too long, and the research goal does not crystallize well. Moreover, a lot of information has been introduced by the authors about the studies already done on this popular plant species.
Paragraph 2.9. Statistical analysis
- “Data were reported as mean ± standard deviation (SD) of at least two independent experiments with three replicates. One-way analysis of variance (ANOVA) was used to analyze the data followed the Least Significant Difference (LSD) test (p<0.05) (XL-Stat-Addinsoft, 230 (2019)”
In the Results section, data from the antioxidant, cytotoxic and antibacterial activities are “reported as mean ± standard deviation”. Where in the Results section are given results from the applied ANOVA statistical analysis methods?
Paragraph 2.5. VOCs analysis
- Are followed any analysis protocol? Which is the literature source? Please, cite it. When this is developed in the present study, the authors must applied data about the validation of the methods.
Paragraph 2.6.2.Cytotoxicity assay
- Are followed any protocol at determination of the cytotoxicity? Which is the literature source? Please, cite it.
- Line 182: “....ten 2-fold dilutions (from 25% to 0.05% v/v) of RH were used and incubated for 48h.”
What solvent is used by the authors to dilute the RH?
Paragraph 3.4.3.TPC and Antioxidant activity
- Line 406-408 “In this case, comparing the values with those found in the bibliography, showed that the effects of the hydrolate in this assay were, in general, greater than other work [86, 90–97]”
How greater? Compared to results obtained to extracts from the red raspberry and obtained using the same FRAP assay method? The differences are due to complex of reasons, including agrometeorological condition, under which the plant are cultivated.
Paragraph 3.2. HS-GC/MS identification of volatile compounds
- In my opinion, it is superfluous to list the species rich in each of the identified substances. It is enough just to indicate their biological activity.The article is not an overview. This will significantly reduce the number of cited sources, and will remain only those related to the studied object or applied methods of analysis.
Conclusion:
- It is not comprehensive. Please, the outhors must mention: the ingredients in the highest amounts; the bacterial strains against which the RH shows good antimicrobial activity; conclusion of the results obtained at the cytotoxic activity tests.
Reviewer 2 Report
Paper seems to be interesting, but some chromatographical issues must be reconsidered.
- Cyclopropylcarbinol is an chromatographical artefact, please re-check the blank or interpretation of MS;
- Cyclobutanemethanol is an chromatographical artefact, please re-check the blank or interpretation of MS;
- 2,3-butanedione could be an artefact, please ensure it's presence; 4. Hotrienol has lit. KI around 1072 (on HP-5 type column); please re-check compound nr 18 in Table 5. b-Myrcene lit. has KI around 991, please re-check compound Table; 6. Limonene has lit. KI around 1025, please re-check compound Table;7. a-terpineol has lit. KI around 1189 , please re-check compound Table;8. Camphor has lit. KI around 1142 , please re-check compound nr Table;9. Methyl salicylate has KI around 1194, please re-check compound in Table 2;10. B-ionone has lit. KI around 1491. please re-check compound in Table 2;11. Re-check also interpretations of other compounds;If possible add as supplementary information crude GC-MS file of selected distillations.Please add info about yield of distillation. Did author performed steam distillation or distillation in water?
Round 2
Reviewer 1 Report
The authors have answered the questions and made the adjustments in accordance with my suggestions. I have no other remarks.